# Real-World Evidence on Adverse Events and Healthcare Resource Utilization in Patients with Chronic Lymphocytic Leukaemia in Spain Using Natural Language Processing: The SRealCLL Study

**DOI:** 10.3390/cancers16234004

**Published:** 2024-11-29

**Authors:** Pau Abrisqueta-Costa, José Antonio García-Marco, Antonio Gutiérrez, José Ángel Hernández-Rivas, Rafael Andreu-Lapiedra, Miguel Arguello-Tomas, Carolina Leiva-Farré, María Dolores López-Roda, Ángel Callejo-Mellén, Esther Álvarez-García, Javier Loscertales

**Affiliations:** 1Haematology Department, Hospital Universitari Vall d’Hebron, 08035 Barcelona, Spain; 2Haematology Department, Hospital Universitario Puerta de Hierro-Majadahonda, 28222 Madrid, Spain; jagarciam@aehh.org; 3Haematology Department, Hospital Son Espases, IdISBa, 07120 Palma de Mallorca, Spain; antoniom.gutierrez@ssib.es; 4Haematology Department, Hospital Universitario Infanta Leonor, Universidad Complutense, 28051 Madrid, Spain; jahernandezr@salud.madrid.org; 5Haematology Department, Hospital Universitario La Fe, 46026 Valencia, Spain; andreu_raflap@gva.es; 6Haematology Department, Hospital de la Santa Creu i Sant Pau, 08041 Barcelona, Spain; marguello@santpau.cat; 7Medical Department, Astrazeneca Farmacéutica Spain S.A., 28050 Madrid, Spain; carolina.leiva@astrazeneca.com (C.L.-F.); angel.callejo@astrazeneca.com (Á.C.-M.); esther.alvarez@astrazeneca.com (E.Á.-G.); 8Haematology Department, Hospital Universitario de la Princesa, 28004 Madrid, Spain; javier.loscertales@salud.madrid.org

**Keywords:** chronic lymphocytic leukaemia, electronic health records, natural language processing, real-world evidence, artificial intelligence, adverse events, healthcare resource utilization

## Abstract

The SRealCLL study collected real-world data on patients with chronic lymphocytic leukaemia (CLL) from Spanish hospitals between 2016 and 2018, focusing on their adverse events (AEs) and healthcare resource use by applying natural language processing to analyse healthcare records. A total of 534 CLL patients were identified, categorizing them into watch and wait (W&W), first-line treatment (1L), and relapse/refractory with second-line treatment (2L) groups. The main antineoplastic treatments were ibrutinib (64.8%) and bendamustine + rituximab (12.6%) in 1L, and ibrutinib (62.1%) and venetoclax (15.5%) in 2L. The study findings revealed that patients in 1L or 2L treatments, representing 43.1% and 10.9% of the cohort, respectively, frequently encountered AEs such as anaemia and thrombocytopenia. These patients also had more outpatient and emergency visits, with almost half of 1L patients requiring hospitalization due to AEs. In conclusion, individuals undergoing 1L or 2L CLL treatments often have heightened healthcare needs, emphasizing the importance of effective and safe management of the disease that optimizes resource utilization, particularly considering the typically older and comorbid nature of this patient population.

## 1. Introduction

Chronic lymphocytic leukaemia (CLL) is one of the most common lymphoid cancers in developed countries, with an incidence of 4–5 per 100,000 inhabitants, being nearly twice as common in men as in women [1]. The median age at diagnosis is 70 years, with the highest number of cases identified in the 65–74-year age group [1,2,3]. Although the disease course and prognosis can vary greatly, the median 5-year survival rate is 88.0% [3].

At diagnosis and in the early stages of the disease, about 80% of patients are asymptomatic [4,5]. The 2018 International Workshop on CLL guidelines recommended against initiating therapy in these patients [4]. Their management follows a “watch-and-wait” (W&W) approach, which includes physical examinations and relevant laboratory tests to monitor disease progression until the balance of risks and benefits favours treatment initiation [4,6]. Some patients may not require therapy for many years, while others experience more aggressive disease with rapid progression, necessitating early treatment [6,7].

Chemo-immunotherapy (CIT) regimens were the standard of care in CLL for many years. However, over the past two decades, significant advances in understanding the pathogenesis of CLL have led to the development of novel targeted therapies that have improved patients’ clinical outcomes. Recent CLL management has shifted the standard of care from CIT to targeted agents, such as Bruton tyrosine kinase inhibitors (ibrutinib, acalabrutinib, zanubrutinib), B-cell lymphoma 2 inhibitors (venetoclax), and new CD20 monoclonal antibodies (obinutuzumab). These targeted agents can be used as first-line treatment (1L) or for relapsed or refractory disease (2L). They have revolutionized the CLL treatment landscape, showing significant improvements in patient survival [8,9,10,11,12,13,14].

The introduction of new therapeutic approaches for CLL has impacted healthcare resource utilization and the economic burden associated with CLL therapy. Both are significant and vary greatly depending on the care setting, treatment regimen, and rate of adverse events (AEs) [15,16,17,18]. While most healthcare utilization by CLL patients is related to diagnostic and therapeutic management, treatment-related AEs also require substantial resources. Several studies have shown that treatment-related AEs are associated with substantial costs, resulting in higher rates of inpatient and emergency department admissions [19,20,21,22]. Therefore, with the increasing number of treatment options for CLL patients, it is crucial to understand treatment patterns, outcomes for novel therapies, and the burden on healthcare systems, particularly in a real-world population [23,24,25,26].

Electronic health records (EHRs) are a valuable real-world data source that can provide relevant clinical data for understanding patient care and fostering cancer research [27,28]. However, the structured data provisioned by many EHR systems are often incomplete, with crucial information such as patient symptoms and outcomes typically recorded as unstructured free text [29,30]. Advances in natural language processing (NLP) and machine learning (ML) techniques have promoted the extraction of clinical information from unstructured texts to fill this information gap, and the application of these techniques in generating real-world evidence (RWE) has been increasing [31,32,33,34,35]. In fact, other studies in the field of onco-hematology have used this innovative technology to process and analyze large volumes of clinical data to provide valuable clinical insights [36,37,38]. However, there are still scarce real-world data related to CLL patients using this novel technology.

The present article is framed within the larger and recently published SRealCLL study, aimed at describing the clinical characteristics and management of patients with CLL in Spain using the unstructured information in EHRs [39]. In this case, our specific goal was to describe the occurrence of AEs and healthcare resource utilization using NLP and artificial intelligence in a real-world scenario.

## 2. Methods

### 2.1. Study Design and Data Source

This was a multicentre, retrospective, observational study of adult patients with CLL. The study was approved by the ethics committee of Hospital Universitario de La Princesa (Madrid, Spain) and conducted in accordance with the Declaration of Helsinki and applicable regulatory requirements. Patient consent was waived due to the study’s retrospective and anonymized data collection.

Data for this study were sourced from EHRs obtained from seven participating hospitals, all of which are academic institutions serving as reference centres in the field of haemato-oncology within the Spanish National Healthcare Network: Hospital Universitario de La Princesa (Madrid), Hospital Universitari Vall d’Hebron (Barcelona), Hospital Universitario Puerta de Hierro-Majadahonda (Madrid), Hospital Universitario Infanta Leonor (Madrid), Hospital Universitari de la Santa Creu i Sant Pau (Barcelona), Hospital Universitario La Fe (Valencia), and Hospital Universitari Son Espases (Mallorca).

### 2.2. Study Population

The study population consisted of adult patients with CLL and available data from 1 January 2016 to 31 December 2018 (Figure 1). We independently evaluated three study groups as follows: (1) *W&W,* CLL patients diagnosed during the study period but without pharmacological treatment detected; (2) first-line treatment (1L), CLL patients with start of first-line treatment detected during the study period; (3) relapse/refractory with second-line treatment (2L), CLL patients with a treatment switch from a 1L to a 2L detected during the study period. The criteria used to classify patients in the different groups are detailed in Figure 1.

In all groups, only CLL patients aged 18 years and older were included. For the 1L group, those patients participating in clinical trials were excluded. However, 2L patients could have participated in clinical trials during their first-line treatments. As shown in Figure 1, due to the inclusion and exclusion criteria, some patients could be included in both the 1L and 2L groups; therefore, while the W&W and 1L groups were mutually exclusive, the 1L and 2L study groups were not.

Patients were included when they met all inclusion and no exclusion criteria for the first time during the study period (index date). Unstructured and structured data were extracted from each individual EHR during a follow-up period (FUP) that ranged from the index date to the last available data or the end of the study period, whichever occurred first. In the 1L group, the treatment switch was also considered the end of the FUP for that stage. As specified before, patients who completed a 1L treatment could later be included in the 2L group.

### 2.3. Extracting Unstructured and Structured Data from EHRs

For this study, all-cause healthcare utilization was evaluated using structured data for hospital outpatient, inpatient, and emergency department visits from the index date to the end of the FUP. Moreover, AEs during the FUP were extracted from unstructured data (free text) and detected by temporally linking events to the treatment period. To address this temporal linking, we discarded events present at the index date. Any outpatient or hospital visit mentioning the included AEs was used to establish a relationship between AEs and these visits, utilizing both unstructured and structured data.

Free text data were extracted from EHR using the EHRead^®^ technology [40,41,42], which applies NLP and ML to free text contained in anonymized EHRs, translating clinical data into medical concepts based on SNOMED-CT terminology. Additional methodology details are described in a recent publication using the same cohort [39]. The EHRead^®^ variable extraction was validated as previously described [43]. Briefly, a set of randomly chosen EHRs was annotated by expert physicians to create a “gold standard”, which was then compared to the EHRead^®^ output for those same EHRs. Each hospital carried out an independent external evaluation to ensure data quality. The EHRead^®^ output was compared to the gold standard using standard metrics including precision, recall, and the harmonic mean of these metrics (F1-score). This validation returned, in most cases, F1-scores ≥ 0.8, which are considered robust detection, indicating that the EHRead^®^ NLP system identified clinical terms adequately. Specific metrics from the external validation are detailed in Appendix A.

### 2.4. Statistical Data Analyses

Descriptive statistics were provided as means and standard deviations (SD) or medians and interquartile range (Q1, Q3) for continuous variables, and percentages for categorical variables, as appropriate. Frequencies of AE and healthcare resource utilization were calculated considering the follow up for each patient as average rates per year. Missing data were handled based on the nature of the data collection process, assuming that physicians recorded clinically relevant information in EHRs. Missing data imputation was performed only for certain dichotomous variables such as AEs or visits, with their absence in EHRs imputed as a true absence (i.e., the patient lacks that AE/visit). No imputation strategies were used for numeric variables. Exploratory p-value calculations between W&W vs. 1L and W&W vs. 2L groups were performed. Data analysis and representation were carried out using “R” software, version 4.0.2 (2020).

## 3. Results

A source population of 385,904 patients was evaluated to determine the study cohort, and 534 were finally included in the study. Among them, 270 (50.6%) underwent W&W, 230 (43.1%) 1L treatment, and 58 (10.9%) 2L treatment during the study period (Figure 1). Twenty-four patients underwent more than one therapeutic strategy during the study period and were included in both the 1L and 2L groups. A general description of patient characteristics and comorbidities has been previously described [39]. Briefly, the median (Q1, Q3) ages of CLL patients in the W&W, 1L, and 2L treatment groups were 75.0 (65.0, 82.0), 75.0 (67.0, 81.0), and 71.0 (61.5, 76.8) years, respectively. In all groups, there was a slightly higher percentage of male patients than female patients, particularly in the 2L treatment group (W&W: 54.8%; 1L: 55.7%; 2L: 63.8%). The most common comorbidities were related to the cardiovascular system and were detected in 43.3% (n = 117) of W&W, 48.3% (n = 111) of 1L, and 51.7% (n = 30) of 2L treatment group patients.

The most common antineoplastic treatment was ibrutinib in 1L (64.8%) and 2L (62.1%). In 1L, it was followed by bendamustine + rituximab (12.6%), obinutuzumab + chlorambucil (5.2%), rituximab + chlorambucil (4.8%), idelalisib + rituximab (3.9%), fludarabine + cyclophosphamide + rituximab (3.5%), ibrutinib + obinutuzumab (2.6%), venetoclax (2.2%), and venetoclax + rituximab (0.4%). In 2L, it was followed by venetoclax (15.5%), idelalisib + rituximab (6.9%), obinutuzumab + chlorambucil (5.2%), bendamustine + rituximab (3.5%), venetoclax + rituximab (3.5%), rituximab + chlorambucil (1.7%), and fludarabine + cyclophosphamide + rituximab (1.7%).

The median (Q1, Q3) follow-up after the diagnosis of CLL was 14.4 (6.0, 22.8) months for the W&W group, 8.4 (2.4, 15.6) months for 1L, and 6 (2.4, 12.0) months for 2L. Loss of follow-up was detected in 15.6% (n = 42) of the W&W group, 28.7% (n = 66) in the 1L group, and 29.3% (n = 17) in the 2L group, respectively (Appendix A).

### 3.1. Occurrence of Adverse Events

A total of 195 patients (72.2%) of the W&W group, 203 patients (88.3%) of the 1L group, and 46 patients (79.3%) of 2L had at least one AE during their FUP. A description of the most frequent AEs after starting CLL regimens across all observed lines of treatment is presented in Figure 2 as the rate of patients with the event per year.

Anaemia, asthenia, fever, and thrombocytopenia were the predominant events in W&W patients and occurred in 0.93, 0.54, 0.49, and 0.42 patients per year, respectively. Several AEs were more commonly seen in groups 1L and 2L compared to W&W, especially those related to cytopenias. The rates of anaemia for W&W, 1L and 2L were 0.93, 2.01, and 2.32, respectively (*p* ≤ 0.05 comparing W&W with both groups), and the rates of thrombocytopenia were 0.42, 1.29, and 1.62, respectively (*p* ≤ 0.05 comparing W&W with both groups). Neutropenia and febrile neutropenia rates were 0.16 and 0.09 for the W&W group, 0.38 and 0.21 for 1L, and 0.57 and 0.47 for the 2L group, respectively (*p* > 0.05). Compared to W&W, major bleeding was more predominant in 1L patients (rate of 0.05 vs. 0.25; *p* ≤ 0.05), who also had more AEs related to the digestive system, such as diarrhea (rate of 0.22 vs. 0.53; *p* ≤ 0.05) or vomiting (rate of 0.12 vs. 0.36; *p* ≤ 0.05), as well as second primary malignancies such as Richter syndrome (rate of 0.01 vs. 0.36; *p* ≤ 0.05) and general AEs such as anorexia (rate of 0.11 vs. 0.41; *p* ≤ 0.05) and unspecified toxicity (rate of 0.07 vs. 0.33; *p* ≤ 0.05). The most common infectious diseases were pneumonia (rates of 0.18 in W&W; 0.38 in 1L; 0.15 in 2L), cytomegalovirus infection (rates of 0.14 in W&W; 0.24 in 1L; 0.11 in 2L), and bacteraemia (rates of 0.10 in W&W; 0.14 in 1L; 0.13 in 2L). Stroke was the only AE which was more frequent in W&W than 1L and 2L (0.31, 0.19 and 0.08, *p* ≤ 0.05 between W&W and 2L). A summary of all events, presented as frequencies and stratified by groups throughout the entire study period, is shown in Appendix A.

### 3.2. Healthcare Services Utilization

The utilization of healthcare resources in CLL patients during the study period across different analysed groups is summarized in Figure 3, Figure 4 and Figure 5. There were no differences between rates of patients with outpatient visits per year (1.67 for W&W, 2.00 for 1L, and 2.40 for 2L; *p* > 0.05). However, the rate of outpatient visits related to AEs was higher in 1L patients compared with W&W (1.07 vs. 0.65, *p* ≤ 0.05) (Figure 3). The rate of hospitalizations (patients with at least one hospitalization per year) was more elevated in both treatment groups (0.88 for W&W, 1.68 for 1L, and 1.9 for 2L, respectively, *p* ≤ 0.05 for the W&W group when compared to 1L group and to 2L group). The rate of hospitalizations related to AEs was also higher in 1L than in W&W (≤0.05) (Figure 3). No differences between groups were found regarding rates of visits to the emergency room, which are shown in Figure 3.

Among patients with outpatient visits, the haematology department was the most visited for all groups. The rate of visits (patients with at least one outpatient visit per year) to this department was lower in the W&W group in comparison with the 1L and 2L groups (0.52, 1.35, and 2.39, *p* ≤ 0.05 for the W&W group when compared to both 1L and 2L groups) (Figure 4). For other outpatient departments, rates were below 0.15 in all the groups, except for cardiology, with a rate of 0.32 for 1L patients, which was higher than that seen in the W&W group (0.11, *p* ≤ 0.05) (Figure 4).

The median (Q1, Q3) hospital length of stay (LOS) was 14.6 (3.9, 32.0) days for W&W patients, 25.1 (7.3, 68.3) days for 1L patients, and 39.5 (19.5, 96.7) days for 2L patients. Moreover, for hospitalizations related to AEs, median (Q1, Q3) LOS was 14.4 (5.0, 31.7) for W&W, 25.5 (8.8, 62) for 1L patients, and 29.8 (10.8, 63.4) days for 2L patients. Hospitalized patients were mostly admitted in the haematology department in all three groups, with higher hospitalization rates per year in the 2L patients, followed by 1L and W&W patients (3.22, 1.50, and 0.56, respectively, *p* ≤ 0.05 for the W&W group when compared to both 1L and 2L groups) (Figure 5). Hospitalization rates in the internal medicine department during the FUP were 0.12 in the W&W group and 0.05 in both the 1L and 2L groups (*p* ≤ 0.05 for the W&W group when compared to the 1L groups). For other departments, no differences were found when comparing hospitalization rates between the W&W group and the 1L and 2L groups.

## 4. Discussion

The SRealCLL study provides a comprehensive analysis of AEs and healthcare resource utilization in a Spanish population of patients with CLL who underwent different therapeutic approaches between 2016 and 2018. The study’s findings highlight that CLL patients frequently experienced hematologic and nonhematologic AEs, and the incidence of these events varied depending on the type of therapeutic approach (W&W, 1L and 2L treatment). Notably, the most common AEs observed in CLL patients were related to the haematological and immune systems. These included anaemia, cytopenia, infectious diseases, and non-major bleeding. Additionally, the study revealed that CLL-treated patients needed more hospitalizations overall and due to AEs, with longer hospital stays than W&W patients.

RWE obtained from Medicare patients in the U.S. also showed that the most frequently recorded AEs in patients receiving 1L or 2L treatment (ibrutinib monotherapy, chlorambucil monotherapy, and bendamustine + rituximab) were neutropenia, hypertension, anaemia, and infection [19]. Our results are aligned with previous reports, as hypertension, anaemia, and bleeding were more frequent in patients under treatment (1L, 2L) than in the W&W group. Moreover, in our study, the incidence of AEs was found to be higher compared to previous findings. Specifically, anaemia and thrombocytopenia in our cohort of patients were detected in 62.6% and 47.8% of 1L-treated patients and 65.5% and 53.4% of 2L-treated patients, but in 34.1% and 21.5% of W&W patients, respectively. However, in the real-world IBRORS-LLC study conducted in Spain, patients treated with single-agent ibrutinib showed incidences of anaemia and thrombocytopenia of 4.8% and 1.2% for 1L and 0.8% and 2.5% for 2L, respectively [23]. In addition, clinical trials such as the Phase Ib/II PCYC-1102 and extension study PCYC-1103 of patients receiving single-agent ibrutinib in 1L or 2L again, the incidences of neutropenia and febrile neutropenia were 18% and 5.5%, respectively [44]. The incidence of major and non-major bleeding shown in our study was also higher in comparison to the incidence reported in other real-world patients [45,46,47]. Thus, Iskierka et al. [45] and Pula et al. [46] reported bleeding-related AEs of any grade in 8.5% and 8.7%, respectively, of patients from Poland with relapse/refractory disease receiving ibrutinib monotherapy, with severe haemorrhagic events in 1.2% and 2.3% of patients. In addition, Ysebaert et al. [47] found that, in patients from France with relapse/refractory disease treated with ibrutinib, the frequency of serious haemorrhagic events was low, occurring in 1.6%. However, they have been widely described, and an integrated analysis of 15 ibrutinib studies (n = 1768 patients) demonstrated a higher proportion of major bleeding on ibrutinib compared with comparator therapy (4.4% vs. 2.8%) in randomized clinical trials (RCTs) [48]. The frequency of atrial fibrillation in our real-world study was 10.0% for 1L and 6.9% for 2L, which is consistent with some data reported from other European real-world studies (6.5%–16%) [36,47]. However, Hillmen et al. reported [26] that 4.6% of patients experienced atrial fibrillation during ibrutinib therapy, and Ghia et al. [49] described an incidence of atrial fibrillation in 5% of patients receiving acalabrutinib monotherapy and in 3% of patients receiving idelalisib plus rituximab or bendamustine plus rituximab. Moreover, in the IBRORS-LLC study, the frequencies of hypertension of any grade and atrial fibrillation were 3.6% and 1.2% for 1L and 4.1% and 3.3% for 2L, respectively [23]. Furthermore, a single-centre study carried out by the Ohio State University’s Comprehensive Cancer Center reported that 71.6% of patients developed de novo hypertension during ibrutinib treatment [50], whereas in the RESONATE and RESONATE-2 clinical trials, 21–26% reported hypertension [51,52]. In this regard, data from our study indicate that hypertension and cardiac arrhythmia, as well as asthenia, fever, stroke, or headache, also happened in the W&W group at the same or higher frequencies. This could be explained by differences in the follow-up period between groups. As mentioned before, the median follow-up period for the W&W group was longer as compared to the 1L and 2L groups, which could explain that patients in this group had more time to develop events. Indeed, when adjusting rates to patients’ follow up, only stroke appeared more frequently in the W&W group than in both the 1L and 2L groups. In addition, even though patients in the SRealCLL study were generally comparable in terms of demographic characteristics and the distribution of baseline comorbidities (hypertension, diabetes mellitus, cardiac arrhythmia, heart failure, and dyslipidaemia being the most common ones), the percentage of patients over 80 years was higher in the W&W group as compared to 1L and 2L (34.8%; 28.7%, and 17.2%) [39].

The higher AE rates observed in our study may be related to our methodology, which was able to extract and analyse a large set of AEs, whereas traditional studies only reported AEs that the physician considered to be remarkable [53,54]. This is compatible with several papers that have exposed the under-reporting of AEs across the field [19,55]. Moreover, it is representative of a patient population that is managed within routine clinical settings, as the methodology employed in this study effectively captures real-world data jotted down by physicians in the patients’ EHRs through the use of NLP. Then, NLP allowed us to identify AEs that were documented in the EHRs but may not have been explicitly reported as such by physicians. Therefore, although the type of AE reported is in line with previous publications, these rates may differ from those observed in RCT, in which the reporting of AEs is carried out prospectively based on clinical assessments rather than retrospectively from EHRs and includes patients who are usually less frail and comorbid. This capability of NLP to recognize and classify events based on unstructured data entries is a significant advantage, as it captures a broader spectrum of clinical information. Finally, it is worth noticing that the study period lasted for two years (2016–2018), leading to a relatively reduced follow-up period, including the beginning of treatment and when the AEs were more prevalent. As a result, it is possible that our study captured the peak occurrence of AEs, which could have subsequently diminished over time, thus explaining the higher apparent incidence compared to findings from other studies. Moreover, it is important to note that some of the studies referenced in this research were conducted during previous time frames. This temporal variation could potentially introduce disparities and reduce their comparability due to differences in the specific treatments employed. For instance, during the study period, second-generation Bruton tyrosine kinase inhibitors such as acalabrutinib or zanubrutinib and the combination of venetoclax plus obinutuzumab were not commercially available or eligible for reimbursement in Spain. This lack of accessibility to these treatments could contribute to the observed differences in AE frequencies. In this sense, second-generation Bruton tyrosine kinase inhibitors reported an improved safety profile, with substantially lower incidence of atrial fibrillation when using zanubrutunib vs. ibrutinib in the ALPINE study [56] and a significantly lower incidence of atrial fibrillation, hypertension, and bleeding, as well as diarrhoea, arthralgia, urinary tract infection, back pain, muscle spasms, and dyspepsia, when using acalabrutinib vs. ibrutinib in the ELEVATE RR study [57].

Concerning healthcare service utilization and its relationship with AEs, our data revealed no differences between the proportion of patients in the W&W group and treatment groups using outpatient resources. However, within the treatment groups, the frequency of outpatient visits per patient was notably higher, primarily driven by the necessity to address AEs. These results are consistent with the established management of W&W patients, as outlined in the current guidelines [58,59]. Typically, these patients are regularly monitored in outpatient visits to evaluate for disease progression, with monitoring including periodic evaluation of blood cell counts and clinical examinations every 3–12 months after the first year [59]. On the other hand, some patients in 1L and 2L could require more frequent outpatient visits to control specific consequences of CLL or AEs. In summary, the analogy in the rates of outpatient visits among W&W patients compared to 1L and 2L is likely due to the need for regular monitoring, while the lower rates due to adverse events reflect the stability of their condition and the absence of treatment-related side effects. To our knowledge, the SRealCLL study provides, for the first time, detailed information about outpatient visits related to different hospital departments. As reported in our study, patients could be visiting the haematology and cardiology departments due to treatment-related AEs such as anaemia, thrombocytopenia, neutropenia, hypertension, and atrial fibrillation. No important differences were seen in emergency room visits. However, it is noteworthy that treated patients, particularly those in 2L, exhibited longer hospital LOS when hospitalization was needed. These differences between 1L and 2L could imply that more effective initial treatment approaches might contribute to cost reduction.

Hospitalizations in all groups were mostly in the haematology department, whilst patients in the W&W group were also admitted to internal medicine departments. Treated patients showed higher rates of cytopenia, which were managed by specific professionals in haematology departments, but W&W patients showed more general problems attended by internal medicine in case of decompensations. The economic burden associated with managing specific AEs associated with commonly used first-line CLL treatment regimens was found to be substantial in the retrospective US claims data analysis from 2005 to 2012 [21]. A separate US claims database analysis of 7965 patients with CLL treated between 2013 and 2015 found that the risk of inpatient admission was increased and related to the number of AEs during CLL treatment. *Kabadi* et al. performed a retrospective cohort study of 7639 patients with CLL who were treated between 2012 and 2015 and demonstrated that the risk of inpatient admission was seven times greater in patients who experienced three to five AEs during first-line CLL treatment. When the healthcare resource use was stratified by the number of AEs, inpatient admissions per patient increased from 3.4% among those with no AEs to 66.1% among those with ≥6 AEs [22]. Other studies also have compared health resource utilization in CLL patients treated with different first-line therapies, showing differences in the resource needs between different treatments [25,60].

The SRealCLL study has several limitations that should be considered when interpreting its findings. First, regulatory policies concerning data confidentiality prevented us from conducting analyses or comparisons between hospitals. Second, the results related to AEs are limited by the availability and accuracy of EHRs, as well as by the information that is collected by physicians in their routine practice, and that ultimately ends up being reflected in the patient’s records. However, there is substantial evidence demonstrating that NLP is highly effective, specifically in adverse event detection; NLP outperforms traditional reporting methods, showing higher sensitivity and specificity [61]. This supports the robustness of our study design and the reliability of our results. Third, due to methodological reasons, assessing the severity of the AEs was not viable, nor were the complications that resulted from them. Thus, we were unable to stratify healthcare resource utilization by AE severity or determine whether it was a significant AE from a specific pharmacological treatment or just related to the pathology. Despite that, we included information about the W&W group, which could work as a control cohort and facilitate understanding of the roles of both the pharmacological treatment, and the pathology, per se. Also, the study period ended in 2018; therefore, we were unable to gather information concerning recent therapeutic drugs in CLL. In our cohort, 64.8% of the patients received the targeted molecule ibrutinib in 1L [39]; hence, the comparison with the most relevant studies in the literature that focus on this drug [23,44,51,52] is feasible and allows us to verify the robustness of the population selected through NLP. Moreover, the descriptive nature of the study also presents certain limitations, such as potential biases and the inability to establish causality. However, it also offers significant benefits, including the ability to analyze a large volume of real-world data over an extended period. Finally, we did not evaluate the specific pharmacological treatment groups, which could entail limitations in the analysis of the described outcomes, and we were unable to perform a post hoc quality assurance on a random sample of our results due to regulatory constraints that ensure data confidentiality.

## 5. Conclusions

In conclusion, the results of the SRealCLL study expand the current evidence on the AEs and healthcare resource utilization in CLL patients in routine clinical practice in Spain by using NLP. Our data reveal that a high percentage of CLL patients in the 1L or 2L groups require healthcare resources, mainly due to AEs, which most commonly include cytopenia such as anaemia and thrombocytopenia. This resource use burden holds huge potential to change the management strategy of CLL patients in order to minimize treatment-related AEs in the outpatient setting and, therefore, reduce preventable emergency department visits and inpatient admissions. These findings support the need to mitigate AEs through effective agents with less off-target effects that may maximize safety with optimized healthcare resource use, which is especially relevant for CLL as it is an indolent chronic disease in older, comorbid patients. Besides this, in the context of an aging population and rising cancer-related healthcare costs, this study could provide relevant findings to estimate healthcare resources demand and, consequently, mitigate healthcare costs.

## Figures and Tables

**Figure 1 cancers-16-04004-f001:**
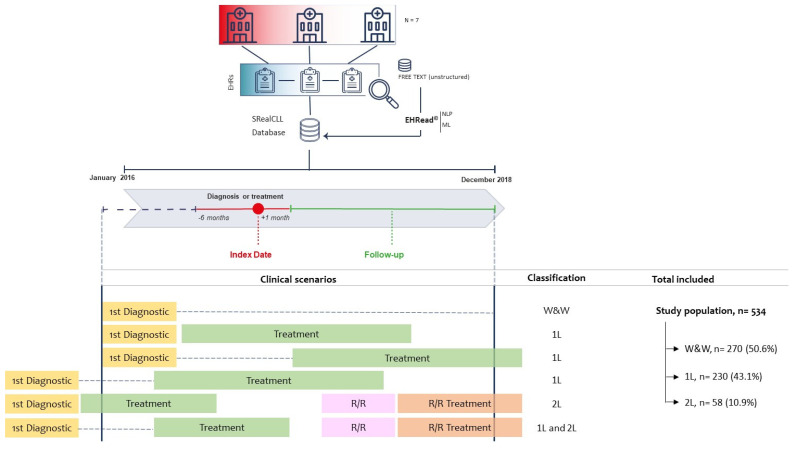
Study design and population. Data were extracted from EHRs corresponding to the study period (from 1 January 2016 to 31 December 2018) from the seven participating hospitals and were analysed using EHRead^®^ technology. The study population (i.e., all patients diagnosed with CLL who fulfil all inclusion/exclusion criteria) comprised 534 patients. Among those, 270 received no treatment throughout the study period (W&W), 230 were in 1L treatment (those with start of first-line treatment detected during the study period), and 58 were in 2L treatment (those with a treatment switch from a 1L to a 2L detected during the study period). Please note that patients in 1L can progress to 2L, such that the sum of the groups is >100%. 1L: first-line; 2L: second-line; CLL: chronic lymphocytic leukaemia; EHRs: electronic health records; ML: machine learning; NLP: natural language processing; W&W: watch and wait; R/R: relapse/refractory.

**Figure 2 cancers-16-04004-f002:**
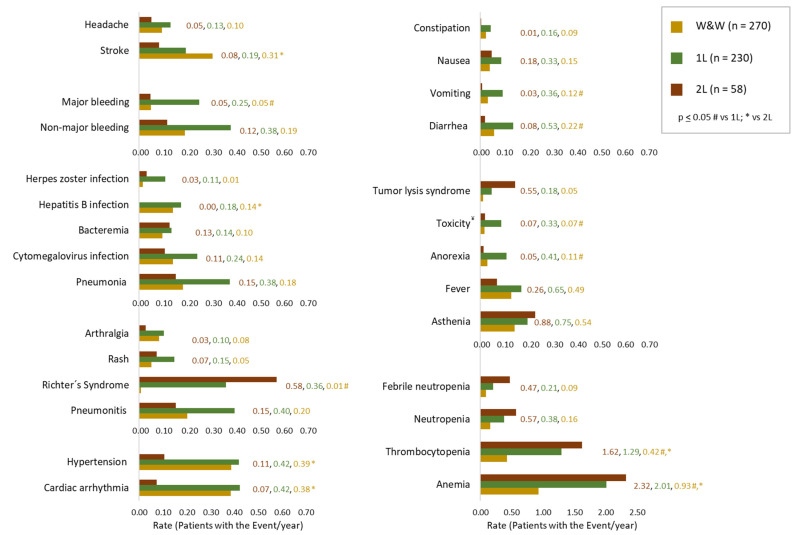
Type and frequency of events and adverse events, showed as rate per year, in patients with CLL under W&W, 1L and 2L treatments. ^¥^ Toxicity was detected as a term itself, without additional information. # *p* ≤ 0.05 W&W vs. 1L; * *p* ≤ 0.05 W&W vs. 2L.

**Figure 3 cancers-16-04004-f003:**
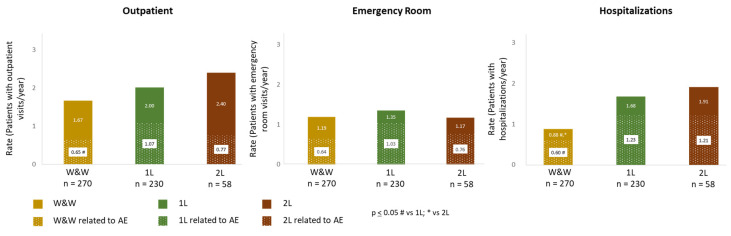
Rates of healthcare resource utilization regarding outpatient, emergency room visits, and hospitalizations in patients with CLL under W&W, 1L, and 2L treatments. The dotted areas indicate rates for those patients who needed these services due to adverse events. # *p* ≤ 0.05 W&W vs. 1L; * *p* ≤ 0.05 W&W vs. 2L.

**Figure 4 cancers-16-04004-f004:**
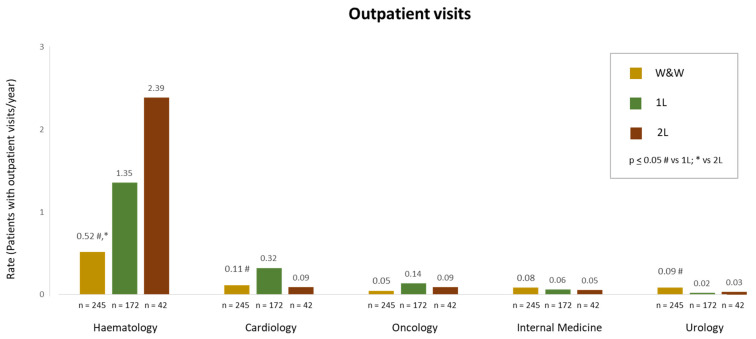
Rates of outpatient visits per year in patients with CLL under W&W, 1L, and 2L treatments. # *p* ≤ 0.05 W&W vs. 1L; * *p* ≤ 0.05 W&W vs. 2L.

**Figure 5 cancers-16-04004-f005:**
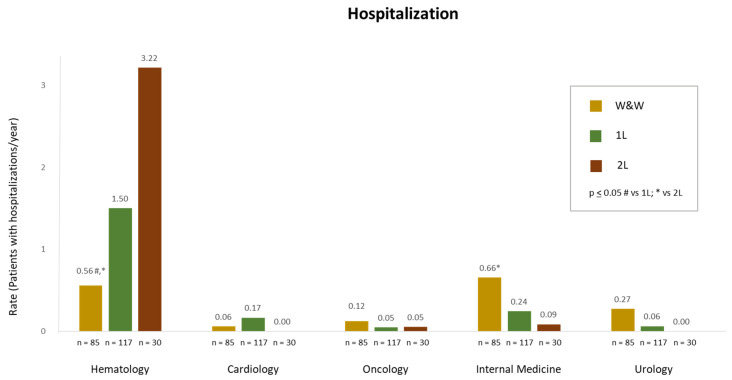
Rates of patients with hospitalizations per year in patients with CLL under W&W, 1L, and 2L treatments. # *p* ≤ 0.05 W&W vs. 1L; * *p* ≤ 0.05 W&W vs. 2L.

## Data Availability

The original contributions presented in the study are included in the article/Appendix A; further inquiries can be directed to the corresponding author.

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
