# Peer review of "Real-World Evidence on Adverse Events and Healthcare Resource Utilization in Patients with Chronic Lymphocytic Leukaemia in Spain Using Natural Language Processing: The SRealCLL Study"

_cancers, 2024, doi:10.3390/cancers16234004_

Round 1

Reviewer 1 Report (Previous Reviewer 3)

Comments and Suggestions for Authors

The authors have responded fully to the questions and suggestions from this reviewer.  

Reviewer 2 Report (Previous Reviewer 1)

Comments and Suggestions for Authors

Thanks for adequately revising the manuscript. I have no more comments. Best regards. 

This manuscript is a resubmission of an earlier submission. The following is a list of the peer review reports and author responses from that submission.

Round 1

Reviewer 1 Report

Comments and Suggestions for Authors

I have read the study by Abrisqueta-Costa et al. with great interest. The authors have utilized natural language processing to analyze adverse events and healthcare resource utilization among CLL patients in Spain. The study provides interesting information, is well conducted, nicely written and the English language is excellent. My only comments are:

1.      Figures need better resolutions and larger word fonts.

2.      For broader impact of the article, the authors should also mention some other studies in hematological and oncological diseases which have used natural language processing, this novel approach, especially studies which have also reported some unexpected observations. (i.e, doi: 10.1002/hem3.143 among others)

3.      Interesting to see increased healthcare utilization also among WW patients: perhaps some of them have been undertreated?

Author Response

Comments and Suggestions for Authors

I have read the study by Abrisqueta-Costa et al. with great interest. The authors have utilized natural language processing to analyse adverse events and healthcare resource utilization among CLL patients in Spain. The study provides interesting information, is well conducted, nicely written and the English language is excellent.

Response: We thank the reviewer for their feedback and these kind words.

My only comments are:

  1. Figures need better resolutions and larger word fonts.

Response: In this revised version of the manuscript, we have enhanced the resolution of the images and increased the font size to improve data visibility. However, given the journal’s template format, we are unsure if these changes are fully reflected in the version reviewed by the reviewers. We are submitting the images independently to ensure they are provided in the highest quality. Should the template affect their display , the journal’s editorial team will have access to the standalone high-resolution images for optimal representation.

  1. For broader impact of the article, the authors should also mention some other studies in haematological and oncological diseases which have used natural language processing, this novel approach, especially studies which have also reported some unexpected observations. (i.e, doi: 10.1002/hem3.143 among others)

Response: We appreciate the reviewer’s valuable comment. We have included a new sentence in the introduction section specifying that other studies in the field of onco-haematology have used this innovative technology to process and analyse large volumes of clinical data, yielding valuable clinical insights. We have included the suggested citation along with other relevant references.

  1. Interesting to see increased healthcare utilization also among WW patients: perhaps some of them have been undertreated?

Response: We thank Reviewer 1 for their comment. Our initial results indicated the percentage of patients who had used healthcare resources (outpatient visits, emergency room visits, or hospitalizations) during their follow-up as well as the number of times they used the resource. However, in response to a very insightful comment from one of the reviewers, we have modified the analyses and adjusted them based on the follow-up time, now showing the results as rates (patients with each resource use per year). Additionally, we have added a comparative exploratory analysis between groups. In this regard, we did not find differences in these rates between patients in the W&W group compared to those in the 1L or 2L groups, although rates of outpatient visits due to AE in 1L were higher than W&W. The observed pattern can be explained by several factors. First, WW patients are typically under strict monitoring protocols to ensure early detection of any disease progression and once detected, a CLL treatment is recommended. This could lead to an important percentage of these patients attending outpatient visits regularly for routine check-ups and monitoring. Second, the lower number of visits overall can suggest that while more WW patients are frequently monitored, they require fewer follow-up visits because their condition remains stable without the need for frequent intervention. Third, the lower percentage of visits due to adverse events among WW patients likely reflects that this group experiences fewer complications or side effects compared to those undergoing first or second-line treatments. This could be due to the fact that WW patients are not exposed to the potential side effects of aggressive treatments, thus reducing the need for visits related to adverse events. Finally, since the WW patients have similar baseline characteristics to those receiving treatment, the differences in visit patterns are likely attributable to the nature of the WW approach itself, rather than differences in patient health status at the outset. In this sense, CLL usually affects to elderly patients with other concomitant comorbidities, that are usually polymedicated. In summary, the analogy in the rates of outpatient visits among WW patients compared to 1L and 2L is likely due to the need for regular monitoring, while the lower rates due to adverse events reflect the stability of their condition and the absence of treatment-related side effects. Based on the Reviewer comment we have strengthened the relevant statement in the discussion section to clarify our points.

Reviewer 2 Report

Comments and Suggestions for Authors

The paper makes a significant contribution to the understanding of CLL management in real-world settings, particularly in the context of AE incidence and healthcare resource utilization. However, the study’s limitations, particularly its retrospective design, lack of AE severity assessment, and potential conflicts of interest, should be carefully considered when interpreting the findings

Author Response

Comments and Suggestions for Authors

The paper makes a significant contribution to the understanding of CLL management in real-world settings, particularly in the context of AE incidence and healthcare resource utilization. However, the study’s limitations, particularly its retrospective design, lack of AE severity assessment, and potential conflicts of interest, should be carefully considered when interpreting the findings

Response: We thank Reviewer 2 for their thorough and insightful review of the manuscript. We are pleased that you agree on the significant contribution that our study makes to the understanding of CLL management in real-world settings, particularly regarding AE incidence and healthcare resource utilization. We fully agree with your observations regarding the study’s constraints. We acknowledge that the retrospective nature of our study presents certain limitations, such as potential biases and the inability to establish causality. However, it also offers significant benefits, including the ability to analyse a large volume of real-world data over an extended period. Additionally, as our study is based on real-life data reported by physicians in their routine clinical practice, there are inherent limitations in the evaluation of certain variables. For instance, the severity of adverse events is typically detailed in clinical trials per protocol, but not consistently in real-world settings. Lastly, we emphasize that all potential conflicts of interest among the authors have been thoroughly disclosed, adhering to necessary and good practice standards. We have reviewed the discussion section, and we have reinforced some of these limitations to ensure that readers can accurately interpret our findings within the appropriate context.

Reviewer 3 Report

Comments and Suggestions for Authors

This paper reports on a real-world data study on patients with chronic lymphocytic leukaemia (CLL) from hospitals in Spain, 2016-2018, to evaluate incidence of adverse events (AEs) and healthcare resource utilization needs.  It would be helpful for the authors to elucidate their goals of this manuscript more clearly:  Is it to purely descriptive of AE incidence and resource utilization?  Or is their study intended to evaluate whether the NLP methodology deployed results in improved detection of AEs?

In the abstract it is stated that AE-related outpatient and ER visits were higher in the 1L vs. W&W and 2L groups; however there is no p value given, were these rates actually significantly higher or just showing a trend?  Similarly, everywhere in the Results section where it states that there were “more” AEs of a particular type, it would be of interest to conduct to statistically test whether there actually was a significantly higher rate of each AE by treatment group.  In the Discussion section they assert that “…the incidence of these events varied depending on the type of therapeutic approach”, and “…CLL treated patients needed more hospitalization”; were these differences statistically significant or not?

However as the authors point out, the median follow-up period was substantially longer for the W&W group than the 1L and 2L groups.  Therefore any comparisons of frequency of AEs would need to be adjust for FUP time, e.g. as a rate per month/year, rather than as a straight percent.  In the authorship list and Author Contributions it is not clear that there was a statistician involved in this work.  Even if it remains a descriptive work, reporting AEs per unit of time would be important to adjust for varying FUP.

It is stated that methodology in this study “likely” enabled us to identify higher rates of AEs that may be underreported using other real-world approaches.  However there are no comparative results presented to support this assertion. It would make the paper more impactful if that were a goal of their study: Does the NLP methodology truly help identify more AEs?  They make comparisons with AEs reported in the literature, however as they acknowledge these other reports involve different timeframes and populations.  In the Discussion the authors state that the higher AE rates observed in their study may be related to using NLP to extract them rather than relying on physician reported AEs.  This assertion is a critical and interesting one, but it is not stated clearly as a goal of their study, and no formal assessment of improvement in AE extraction is given.

The authors describe utilizing a gold standard in the development of their NLP queries, however was there any post-hoc QA applied to a random sample of their results to see if a similar level of precision and recall were attained, i.e. how do we know we can rely on their results as accurate and representative of the incidence of AEs in the population studied? 

It is a plus that the study was multi-centered, and the methodology applied appears to be applicable to the 7 different EHR systems.  However they state that comparisons among the EHRs were not allowed under the study regulations.  It would have been interesting to see whether there were there any differences in AE detection via application of NLP across the 7 centers.

It is stated that there was some crossover from the 1L to the 2L groups, but couldn’t there also have been transitions from W&W to 1L?  

Although publication (36) contains the detailed methodology and metrics for the authors’ NLP and ML approach, it would be helpful to the readers of this article to include a bit more information in the current manuscript, e.g. the precision, recall, and F score results.

Comments on the Quality of English Language

In Keywords, the term “advert events” should be “adverse events”.

In the abstract “AEs-related” should be “AE-related”.   It would help to add the word “groups” before the parenthetical phrase with the rates, i.e. “…in the IF compared to W&W and 2L groups (….”.

In the Introduction, 2nd paragraph, the phrase “in the early stages” should be moved up, i.e. “At diagnosis and in the early stages, about 80%....”.     Remove the word “in” in the phrase: “….while in others experience..”.

In Methods, 2.2, for clarity perhaps reword the following:  “…first-line treatment starting detected during the study period;” to “…start of first-line treatment detected during the study period”.  

Similarly in Figure 1 legend suggest rewording to  “…230 were in IL treatment (those with start of first-line treatment detected during the study period),”.

Reword “treatment switch was also considered” to “treatment switch also was considered”.

In 2.3, “To address this” should be “To address this temporal linking, “.  Change to “Free text data were extracted”, rather than “was extracted”.

In 2.4, reword to “.., with their absence in EHRs imputed as a true absence”, removing the word “was”.

In 3.1 I believe it should be singular for “…at least one AE during their FUP.” (rather than AEs).  Conversely I believe it should be AEs (plural) in “Several AEs were more commonly seen…”.

In the legends for Figures 2 and 3, should “LLC” be “CLL”?

For clarity perhaps reword the beginning of the Results to:  A source population of 385,904 patients was evaluated to determine the study cohort, and 534 were finally included in the study.”, rather than the population being “analysed”.

In Results, reword “…number of visits to the haematology department was also higher…” to “also was higher”.

Change “..the 2L patients had more median visits..” to “the number of median visits was higher in the 2L patients”, and “This trend was also observed…” to “also was observed”, and make it AEs (plural) in “…hospitalizations related to AEs (Figure 3B).”

Change “being” to “with” in “…with 2L patients requiring the most hospital admissions…”.

Reword “..bleeding shown in our study was also higher” to “also was higher”.

I believe “of” should be “in” for “in Polish relapse/refractory patients..”.

Change to “data from our study indicate” rather than “indicates”.

In the Discussion the word “who” is missing from “…and including patients who are usually less frail and comorbid.”

Change “Other studies have also compared..” to “also have compared”.

It is not clear what is being said in “ and allows us to objective the plausibility of the population selected through NLP”?

Change “Our data concludes” to “Our data demonstrate…” and “high percentage ….require healthcare resources” rather than “requires”.

Author Response

Comments and Suggestions for Authors

This paper reports on a real-world data study on patients with chronic lymphocytic leukaemia (CLL) from hospitals in Spain, 2016-2018, to evaluate incidence of adverse events (AEs) and healthcare resource utilization needs.  It would be helpful for the authors to elucidate their goals of this manuscript more clearly:  Is it to purely descriptive of AE incidence and resource utilization?  Or is their study intended to evaluate whether the NLP methodology deployed results in improved detection of AEs?

Response: The present article is framed within the larger and recently published SRealCLL study, aimed at describing the clinical characteristics and management of patients with CLL in Spain, using the unstructured information in electronic health records (EHRs). The specific goal with this report was to describe the occurrence of AEs and healthcare resource utilization by management (WW, 1L or 2L) using NLP and artificial intelligence in a real-world scenario. While as suggested by the Reviewer, the potential for improved detection of AEs using Natural Language Processing (NLP) is an interesting avenue, it is not within the scope of our current study. To evaluate whether the NLP methodology results in improved detection of AEs, a comparative study involving manual data extraction from EHRs would be necessary. This would provide a direct comparison between NLP-based and manual extraction methods. Unfortunately, such a comparative analysis is beyond the capabilities of our current dataset and study design. However, it is worth noting that numerous studies have demonstrated the efficacy of NLP in extracting clinical data with higher accuracy and fewer missing values compared to traditional methods. For instance, NLP has been shown to effectively extract cancer-related concepts from clinical notes with high accuracy and sensitivity. Additionally, in adverse event detection, NLP outperforms traditional reporting methods, showing higher sensitivity and specificity. These studies suggest that NLP methodologies generally result in more comprehensive and accurate data extraction, which supports the reliability of our findings.

In summary, our study was purely descriptive of AE incidence and resource utilization, and it did not intend to evaluate whether the NLP methodology deployed results in improved detection of AEs. While our study does not aim to evaluate the NLP methodology itself in terms of improved detection of AEs, existing literature supports the notion that NLP can enhance data extraction accuracy and completeness. This provides a robust foundation for our descriptive analysis. We have included this concept in the discussion section of the revised manuscript.

In the abstract it is stated that AE-related outpatient and ER visits were higher in the 1L vs. W&W and 2L groups; however, there is no p value given, were these rates actually significantly higher or just showing a trend?  Similarly, everywhere in the Results section where it states that there were “more” AEs of a particular type, it would be of interest to conduct to statistically test whether there actually was a significantly higher rate of each AE by treatment group.  In the Discussion section they assert that “…the incidence of these events varied depending on the type of therapeutic approach”, and “…CLL treated patients needed more hospitalization”; were these differences statistically significant or not?

Response:  Thank you for your insightful comment. As explained before, the SRealCLL study was designed as a descriptive analysis of patients with CLL in Spain, stratified based on the therapeutic management received such as WW, 1L, or 2L treatments (see previous publication in Loscertales J et al. Cancers 2023, 15(16): 4047). This descriptive approach allowed for patients to be included in both the 1L and 2L groups, resulting in potential duplication. Consequently, comparing these groups directly does not meet the independence assumption required for statistical analysis. However, we understand your concern and have now included p-values for exploratory comparisons between the WW vs. 1L and WW vs. 2L groups, which can be compared independently.

Additionally, following your next recommendation, we have conducted these comparative exploratory analyses not based on the percentage of patients but by adjusting the results for the follow-up period of individuals, presenting them as rates per year. These analyses have been included in the revised manuscript. We also have updated the Methods section to explain these new analyses and have reorganized the Results section to reflect the new figures. Furthermore, we have adjusted the wording in the Results and Discussion sections to ensure rigor and accuracy in reporting our findings.

However, as the authors point out, the median follow-up period was substantially longer for the W&W group than the 1L and 2L groups. Therefore, any comparisons of frequency of AEs would need to be adjust for FUP time, e.g. as a rate per month/year, rather than as a straight percent.  In the authorship list and Author Contributions it is not clear that there was a statistician involved in this work.  Even if it remains a descriptive work, reporting AEs per unit of time would be important to adjust for varying FUP.

Response: We thank the reviewer for their feedback in this analysis and we agree with the rationale proposed. In consequence, all percentages previously included in the Figures 2, 3, 4 and 5 have been replaced in the manuscript considering the follow-up for each patient, with the average rates per year. All statistical analyses were performed by the Savana Research Group, as indicated in the acknowledgement section.

It is stated that methodology in this study “likely” enabled us to identify higher rates of AEs that may be underreported using other real-world approaches.  However, there are no comparative results presented to support this assertion. It would make the paper more impactful if that were a goal of their study: Does the NLP methodology truly help identify more AEs?  They make comparisons with AEs reported in the literature, however as they acknowledge these other reports involve different timeframes and populations.  In the Discussion the authors state that the higher AE rates observed in their study may be related to using NLP to extract them rather than relying on physician reported AEs.  This assertion is a critical and interesting one, but it is not stated clearly as a goal of their study, and no formal assessment of improvement in AE extraction is given.

Response: We appreciate the reviewer’s insightful comments. Evaluating whether the NLP methodology truly helps to identify more AEs is indeed an interesting and valuable objective. However, the primary goal of the SRealCLL study, as approved by the ethics committee, was to describe clinical characteristics and management of CLL patients using NLP and artificial intelligence. Evaluating the comparative effectiveness of NLP in detecting AEs was not within the scope of our initial study design and in consequence we do not have the corresponding approval to develop it. Moreover, the efficacy of NLP in enhancing data extraction accuracy and completeness has been well-documented in previous studies. Numerous publications have demonstrated that NLP can identify clinical events that may be underreported or missed by traditional methods. Therefore, while our study did not aim to formally assess this, we leveraged the established strengths of NLP to ensure robust data extraction.

Regarding the higher AE rates observed in our study, we stated that this may be attributed to the NLP methodology. NLP allowed us to identify AEs that were documented in the EHRs but may not be explicitly reported as such by physicians. This capability of NLP to recognize and classify events based on unstructured data entries is a significant advantage, as it captures a broader spectrum of clinical information. We agree with the Reviewer 3 that further research should be addressed in this regard. We have adapted the discussion section in the revised manuscript to clarify our study’s scope and the rationale behind our methodology.

The authors describe utilizing a gold standard in the development of their NLP queries, however was there any post-hoc QA applied to a random sample of their results to see if a similar level of precision and recall were attained, i.e. how do we know we can rely on their results as accurate and representative of the incidence of AEs in the population studied? 

Response: Thank you for your insightful question. We understand the importance of validating our NLP queries to ensure the accuracy and representativeness of our results. However, due to certain limitations, we were unable to perform a post-hoc quality assurance (QA) on a random sample of our results for this study. The statistical analysis included in this study was carried out by Savana Research S.L. based on an anonymized database to ensure data confidentiality. The anonymization process included a first pseudonymization process that took place in the centres participating in the research, followed by an anonymization process that took place in Medsavana, who is also responsible for ingesting, integrating, and processing the data for variable extraction using NLP technology. After that, neither Savana Research nor its employees, nor the principal investigators, had access to the personal data information as defined in the applicable data protection regulations, and they did not have the possibility of reversing the anonymization carried out. Given that Savana Research S.L. had no access to the EHRs and that patient IDs were anonymized, we were unable to perform a direct validation on the original dataset. However, we acknowledge the importance of this validation and have updated the limitations section in the manuscript to better reflect these findings. We hope the reviewer will appreciate our efforts to ensure data confidentiality and the challenges associated with patient-level outcome validation in real-world evidence studies.

It is a plus that the study was multi-centered, and the methodology applied appears to be applicable to the 7 different EHR systems.  However, they state that comparisons among the EHRs were not allowed under the study regulations.  It would have been interesting to see whether there were there any differences in AE detection via application of NLP across the 7 centres.

Response: We appreciate the reviewer’s observation regarding the multi-centred nature of our study and the applicability of our methodology across the seven different EHR systems. We agree that it would have been interesting to analyse potential differences in AE detection via NLP across the various centres. Such an analysis could provide valuable insights into the consistency and reliability of NLP applications in diverse clinical settings.

However, due to regulatory constraints, we were unable to perform comparisons among the EHR systems. Specifically, we are required to ensure data anonymization to comply with ethical standards and to be exempt from obtaining informed consent. To guarantee the anonymization of the data, we must work with aggregated data. Individualized data by hospital would only allow us to ensure pseudonymization, which does not meet the stringent requirements for data protection and privacy.

Therefore, while we acknowledge the potential value of such an analysis, our study design and regulatory obligations necessitated the use of aggregated data to maintain the highest standards of data privacy and ethical compliance.

It is stated that there was some crossover from the 1L to the 2L groups, but couldn’t there also have been transitions from W&W to 1L?  

Response: Thank you for your insightful question. To clarify, any transition from W&W to 1L during the study period was indeed considered as 1L. Patients in the W&W group were defined as those diagnosed during the study period but without any pharmacological treatment detected throughout the entire period. Conversely, a patient with the initiation of treatment detected during the study period was classified as 1L. The definition of 2L included relapsed patients who started second-line treatment during the study period, which allowed for the inclusion of patients meeting both 1L and 2L criteria within the study period, as these were not mutually exclusive conditions. However, the definitions for W&W and 1L were mutually exclusive. Therefore, there could not have been transitions from W&W to 1L without reclassifying the patient as 1L. We have adapted the paper (methods section) to clarify this aspect more accurately.

Although publication (36) contains the detailed methodology and metrics for the authors’ NLP and ML approach, it would be helpful to the readers of this article to include a bit more information in the current manuscript, e.g. the precision, recall, and F score results.

Response: We agree that the metrics results are relevant and should be visible to the readers. Initially, we did not include these results in the manuscript because they were already published in the context of a previous publication with the same cohort of patients, and we wanted to avoid duplicating results. However, we recognize the importance of these metrics in providing credibility to our findings. Therefore, we have adapted the manuscript to include the precision, recall, and F-score results as supplementary material. This addition will help readers better understand and evaluate the performance of our NLP and ML approach.

Comments on the Quality of English Language

In Keywords, the term “advert events” should be “adverse events”.

Response: We thank the reviewer for this comment, we have corrected the indicated typo.

In the abstract “AEs-related” should be “AE-related”.   It would help to add the word “groups” before the parenthetical phrase with the rates, i.e. “…in the IF compared to W&W and 2L groups (….”.

Response: We have followed the reviewer suggestion in this new manuscript version.

In the Introduction, 2nd paragraph, the phrase “in the early stages” should be moved up, i.e. “At diagnosis and in the early stages, about 80%....”.     Remove the word “in” in the phrase: “….while in others experience..”.

Response: We have followed the reviewer suggestion in this new manuscript version.

In Methods, 2.2, for clarity perhaps reword the following:  “…first-line treatment starting detected during the study period;” to “…start of first-line treatment detected during the study period”. Similarly in Figure 1 legend suggest rewording to  “…230 were in IL treatment (those with start of first-line treatment detected during the study period),”.

Response: Thanks for the comment, we have adapted the text as suggested.

Reword “treatment switch was also considered” to “treatment switch also was considered”.

Response: It has been modified, thanks.

In 2.3, “To address this” should be “To address this temporal linking, “.  Change to “Free text data were extracted”, rather than “was extracted”.

Response: It has been modified, thanks.

In 2.4, reword to “.., with their absence in EHRs imputed as a true absence”, removing the word “was”.

Response: It has been modified, thanks.

In 3.1 I believe it should be singular for “…at least one AE during their FUP.” (rather than AEs).  Conversely I believe it should be AEs (plural) in “Several AEs were more commonly seen…”.

Response: Suggestions have been implemented, thanks.

In the legends for Figures 2 and 3, should “LLC” be “CLL”?

Response: Thanks for noting this mistake, it has been corrected.

For clarity perhaps reword the beginning of the Results to:  A source population of 385,904 patients was evaluated to determine the study cohort, and 534 were finally included in the study.”, rather than the population being “analysed”.

Response: Suggestions have been implemented, thanks.

In Results, reword “…number of visits to the haematology department was also higher…” to “also was higher”.

Response: It has been corrected, thanks.

Change “..the 2L patients had more median visits..” to “the number of median visits was higher in the 2L patients”, and “This trend was also observed…” to “also was observed”, and make it AEs (plural) in “…hospitalizations related to AEs (Figure 3B).”

Response: The suggestions have been applied, thanks.

Change “being” to “with” in “…with 2L patients requiring the most hospital admissions…”.

Response: It has been done, thanks.

Reword “..bleeding shown in our study was also higher” to “also was higher”.

Response: It has been done, thanks.

I believe “of” should be “in” for “in Polish relapse/refractory patients..”.

Response: It has been changed, thanks.

Change to “data from our study indicate” rather than “indicates”.

Response: It has been modified, thanks.

In the Discussion the word “who” is missing from “…and including patients who are usually less frail and comorbid.”

Response: It has been addressed as suggested, thanks.

Change “Other studies have also compared..” to “also have compared”.

Response: It has been done, thanks.

It is not clear what is being said in “ and allows us to objective the plausibility of the population selected through NLP”?

Response: We tried to specify that by comparing these results, we can confirm that the chosen population by the NLP is appropriate and reasonable. We have reworded this sentence changing “… allows us to objective the plausibility of the population selected through NLP” by “… allows us to verify the robustness of the population selected through NLP”.

Change “Our data concludes” to “Our data demonstrate…” and “high percentage ….require healthcare resources” rather than “requires”.

Response: These comments have been corrected, thanks.
